# CascadeFormer: A Family of Two-stage Cascading Transformers for Skeleton-based Human Action Recognition

## Abstract

Skeleton-based human action recognition leverages sequences of human joint co-ordinates to identify actions performed in videos. Owing to the intrinsic spatiotemporal structure of skeleton data, Graph Convolutional Networks (GCNs) have been the dominant architecture in this field. However, recent advances in transformer models and masked pretraining frameworks open new avenues for representation learning. In this work, we propose **CascadeFormer**, a family of two-stage cascading transformers for skeleton-based human action recognition. Our framework consists of a masked pretraining stage to learn generalizable skeleton representations, followed by a cascading fine-tuning stage tailored for discriminative action classification. We evaluate CascadeFormer across three benchmark datasets—Penn Action, N-UCLA, and NTU RGB+D 60—achieving competitive performance on all tasks. To promote reproducibility, we will release our code and model checkpoints.

## 1 Introduction

Human action recognition is a fundamental computer vision task that aims to identify the action performed by a target person in a given video, represented as a sequence of frames. Human actions can be interpreted through various modalities, such as raw RGB images Zhang et al. (2021), skeletal joint coordinates Zhao et al. (2019), or a fusion of both Kim et al. (2023). Compared to RGB-based approaches, skeleton-based action recognition offers distinct advantages Zhou et al. (2023): it is computationally efficient due to relying solely on human joint coordinate data, and it is more robust to environmental noise and variations in camera viewpoints. Using these strengths, skeleton-based action recognition has been explored through various methods, including probabilistic models Zhao et al. (2019), recurrent neural networks (Shahroudy et al., 2016; Liu et al., 2016), and graph convolutional networks (GCN) Yan et al. (2018). Nevertheless, the rapid advancement of attention mechanisms Vaswani et al. (2023) in natural language processing has driven the widespread adoption of transformer architectures in computer vision. The success of vision transformers Dosovitskiy et al. (2021); Rao et al. (2021) across tasks such as image classification Ridnik et al. (2021), multimodal learning Radford et al. (2021), and visual instruction tuning Liu et al. (2023) underscores the potential of transformers for skeleton-based action recognition. Recent approaches incorporate novel attention designs tailored to spatiotemporal data Bertasius et al. (2021); Zhou et al. (2023); Wang et al. (2023); however, these transformer-based models are predominantly trained in an **end-to-end** manner.

However, instead of relying on end-to-end training, which may risk overfitting on downstream tasks, both language modeling Devlin et al. (2019) and multimodal learning Srivastava & Sharma (2023; 2024) have widely embraced masked pretraining as a precursor to supervised adaptation. ***Motivated by the goal of equipping simple transformers with masked pretraining***, we propose **CascadeFormer**, a family of two-stage cascading transformers for skeleton-based human action recognition. CascadeFormer comprises a lightweight transformer for masked pretraining and an additional transformer for fine-tuning on action labels. In this paper, we begin with a review of prior work in masked pretraining with transformers and skeleton-based action recognition, followed by a brief overview of the input skeleton data format. We then present our key contributions:

1. We introduce **CascadeFormer**, a framework comprising three model variants that share a unified pipeline involving masked pretraining followed by cascading fine-tuning.

2. We conducted extensive evaluations of CascadeFormer on three widely used and diverse datasets: Penn Action, N-UCLA, and NTU RGB+D 60.

3. We performed extensive ablation studies, summarized in the main paper and further expanded in the Appendix, to analyze the impact of architectural choices and training configurations.

Finally, we conclude with a discussion of the implications of our work.

## 2 RELATED WORK

In this section, we first provide a brief overview of the development of masked pretraining frameworks in various fields. We then review widely adopted approaches in the recognition of human action based on skeletons.

**Masked Pretraining Framework**   The masked pre-training was first introduced in the domain of language modeling, most notably with BERT Devlin et al. (2019). By randomly masking $15\%$ of the input tokens and training the model to predict them in a self-supervised manner, BERT can learn deep bidirectional representations Devlin et al. (2019). A similar paradigm has been extended to computer vision with masked autoencoders (MAEs) He et al. (2021), which apply a significantly higher masking ratio (up to $80\%$), enabling effective self-supervised pretraining for vanilla vision transformers Dosovitskiy et al. (2021). More recently, the masked pretraining framework has been rapidly adopted in multimodal learning. Models such as OmniVec Srivastava & Sharma (2023) and OmniVec2 Srivastava & Sharma (2024) take advantage of masked auto-encoding techniques to learn unified representations in diverse modalities, including image, text, video, and audio.

**Skeleton-based Action Recognition**   Human action recognition can be explored through various modalities, including raw RGB images Zhang et al. (2021), skeletal joint coordinates Zhao et al. (2019), or a fusion of both Kim et al. (2023). In contrast to RGB-based methods, **skeleton**-based action recognition is characterized by its computational efficiency, owing to its reliance solely on human joint coordinate data, and its robustness to environmental noise and camera viewpoint variations, since it preserves only the spatiotemporal information of keypoints Zhou et al. (2023). Although probabilistic models Zhao et al. (2019) demonstrated promising performance, deep learning approaches have become dominant in this domain. Given the temporal nature of video sequences, recurrent neural networks (RNN) Shahroudy et al. (2016); Liu et al. (2016) have proven effective in modeling temporal dynamics. Furthermore, convolutional neural networks (CNNs) Cai et al. (2023); Duan et al. (2022) have been adapted to this task by transforming skeleton data into pseudo-images, allowing convolutional operations to be applied. Furthermore, graph-based neural networks, including graph neural networks (GNNs) Shi et al. (2019) and graph convolutional networks (GCNs) Yan et al. (2018), have gained popularity due to the natural graph structure of human skeletons, where joints and bones are represented as vertices and edges, respectively. More recently, transformer architectures Vaswani et al. (2023) have attracted increasing attention in skeleton-based action recognition. These models are often enhanced with efficient spatiotemporal attention mechanisms Zhou et al. (2023); Do & Kim (2024); Bertasius et al. (2021); Wang et al. (2023), and are typically trained in an end-to-end fashion.

## 3 PRELIMINARIES

**Skeleton Data**   Skeletons, or pose maps, refer to sets of Cartesian coordinates representing the key joints of the human body. Formally, given a batch of $B$ video sequences, each consisting of $T$ frames, with $J$ joints per frame in a $C$-dimensional space, the input skeleton data $X$ is typically structured as: $X \in \mathbb{R}^{B \times C \times T \times J}$. Although certain datasets such as Penn Action define skeletons in a two-dimensional space ($C = 2$), the most recent data sets—including N-UCLA and NTU RGB+D 60—represent joint coordinates in a three-dimensional space ($C = 3$) to more accurately capture 3D motion. Since video lengths may vary across samples, it is common to either sample a fixed number of frames or apply dynamic padding based on the longest sequence in the batch.

**Multi-person Skeleton Data** Some datasets, such as NTU RGB+D 60, include actions involving multiple individuals. In these cases, assuming up to $M$ persons appear in each video, the input skeleton data $X$ can be extended to: $X \in \mathbb{R}^{B \times C \times T \times J \times M}$. When multiple persons are present, many approaches Do & Kim (2024); Zhou et al. (2023) encode all detected individuals. However, some methods Shahroudy et al. (2016) opt to retain only the most active person in the scene to reduce computational complexity or avoid noise from irrelevant actors.

# 4 CASCADEFORMER

To equip vanilla transformers with masked pretraining, a technique widely adopted in language modeling and multimodal learning, we propose **CascadeFormer**, a family of two-stage cascading transformers for skeleton-based human action recognition. As illustrated in Figure 1, CascadeFormer employs a lightweight transformer for masked pre-training, which is then followed by an additional transformer to fine-tune the action labels (Figure 2). We design a family of three model variants that differ in their feature extraction modules, as shown in Figure 3. The architecture is composed of three key components: masked pre-training, cascading fine-tuning, and feature extraction, each of which is described in detail in the following subsections.

## 4.1 MASKED PRETRAINING COMPONENT

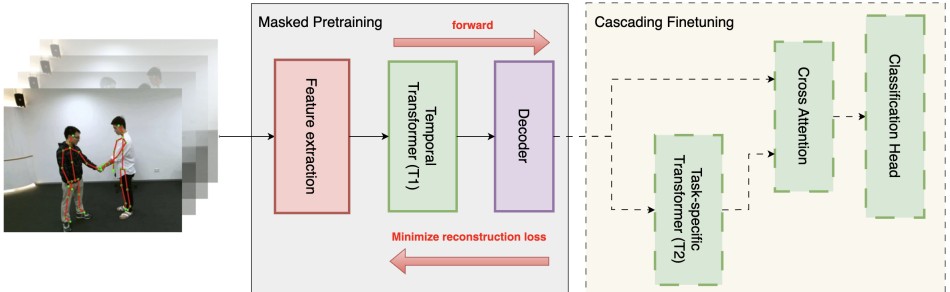

Figure 1: **Overview of the masked pretraining component in CascadeFormer.** A fixed percentage of joints are randomly masked across all frames in each video. The partially masked skeleton sequence is passed through a feature extraction module to produce frame-level embeddings, which are then input into a temporal transformer (T1). A lightweight linear decoder is applied to reconstruct the masked joints, and the model is optimized using mean squared error over the masked positions. This stage enables the model to learn generalizable spatiotemporal representations prior to supervised finetuning.

The masked pretraining component, illustrated in Figure 1, enables the model to learn generalizable spatiotemporal dependencies in a self-supervised manner prior to any supervised training on action labels. Inspired by masked modeling strategies in other domains, we apply a joint-level masking scheme: specifically, 30% of the joints across all frames are randomly selected and masked by setting their coordinates to zero, while the remaining 70% retain their original Cartesian values. The partially masked skeleton sequence is then passed through a feature extraction module to generate frame-level embeddings. Although the three variants of CascadeFormer differ in their feature extraction designs, they all follow a common principle: each frame is treated as a token and projected into a high-dimensional embedding space. Formally, given a batch of $B$ video sequences, each containing $T$ frames with $J$ joints per frame in a $C$-dimensional space, the resulting embedding tensor is denoted as $E \in \mathbb{R}^{B \times T \times \text{embed\_dim}}$, where embed_dim denotes the transformer embedding dimension. These frame embeddings are then input into a vanilla temporal transformer backbone (T1), as shown in Figure 1. The output of T1, denoted as $E_{\text{pretrain}}$, maintains the same shape: $E_{\text{pretrain}} \in \mathbb{R}^{B \times T \times \text{embed\_dim}}$. To reconstruct missing joint information, a lightweight linear decoder is applied to the output of T1. This decoder is used exclusively during pretraining and is discarded during downstream fine-tuning. The model is trained to minimize the reconstruction error only in the masked joints. Let $masked\_X \in \mathbb{R}^{B \times C \times T \times J}$ represent the masked input skeletons and

$masked\_X' \in \mathbb{R}^{B \times C \times T \times J}$ be the reconstructed output. The loss function is defined as the mean squared error (MSE) over the masked joints:

$$\mathcal{L}_{\text{MSE}} = \|masked\_X - masked\_X'\|^2$$

## 4.2 CASCADING FINETUNING COMPONENT

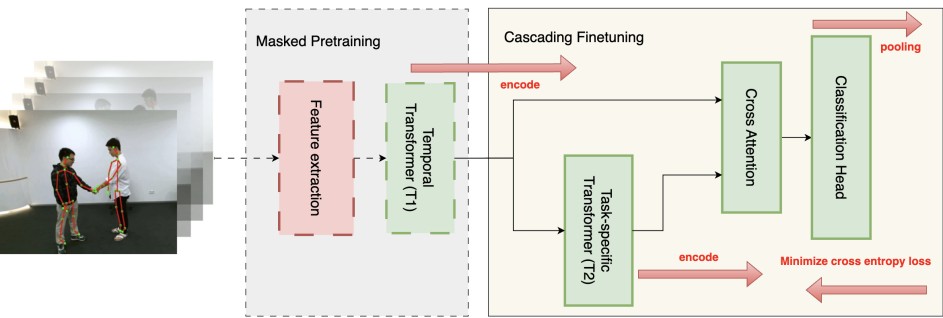

Figure 2: **Overview of the cascading finetuning component in CascadeFormer.** The frame embeddings produced by the pretrained temporal transformer backbone (T1) are passed into a task-specific transformer (T2) for hierarchical refinement. The output of T2 is fused with the original embeddings via a cross-attention module. The resulting fused representations are aggregated through frame-level average pooling and passed to a lightweight classification head. The entire model—including T1, T2, and the classification head—is optimized using cross-entropy loss on action labels during finetuning.

After masked pretraining, we introduce a ***cascading*** finetuning stage, as illustrated in Figure 2. This design is inspired by the hierarchical adaptation strategy proposed in OmniVec2 Srivastava & Sharma (2024) for multimodal learning. Unlike conventional fine-tuning, where a lightweight classification head is attached directly to a pre-trained transformer, CascadeFormer incorporates an additional task-specific transformer (T2) to refine features in a hierarchical manner.

Specifically, the frame embeddings $E \in \mathbb{R}^{B \times T \times \text{embed\_dim}}$ obtained from the pretrained backbone (T1) are passed into the task-specific transformer (T2), which maps them into a new embedding space of the same dimensionality: $E_{\text{finetune}} \in \mathbb{R}^{B \times T \times \text{embed\_dim}}$. These refined embeddings $E_{\text{finetune}}$ are then fused with the original pretrained embeddings $E_{\text{pretrain}}$ via a cross-attention module, as shown in Figure 2. Following the standard attention mechanism Vaswani et al. (2023), cross attention is computed as:

$$\text{Attention}(Q, K, V) = \text{softmax}\left(\frac{QK^\top}{\sqrt{d}}\right) V$$

$$E_{\text{cross}} = \text{Attention}(E_{\text{pretrain}}, E_{\text{finetune}}, E_{\text{finetune}})$$

To obtain a fixed-size video-level representation, we apply frame-level average pooling over the cross-attended embeddings:

$$E_{\text{avg}} = \frac{1}{T} \sum\nolimits_{t=1}^{T} E_{\text{cross}}[:, t, :] \in \mathbb{R}^{B \times \text{embed\_dim}}$$

Finally, a lightweight classification head is applied to the pooled embeddings, and the model is optimized using the standard cross-entropy loss with respect to the ground-truth action labels. All parameters of the pre-trained transformer backbone (T1), as well as the newly introduced T2 module and classification head, are trainable during the cascading fine-tuning stage.

## 4.3 FEATURE EXTRACTION MODULE

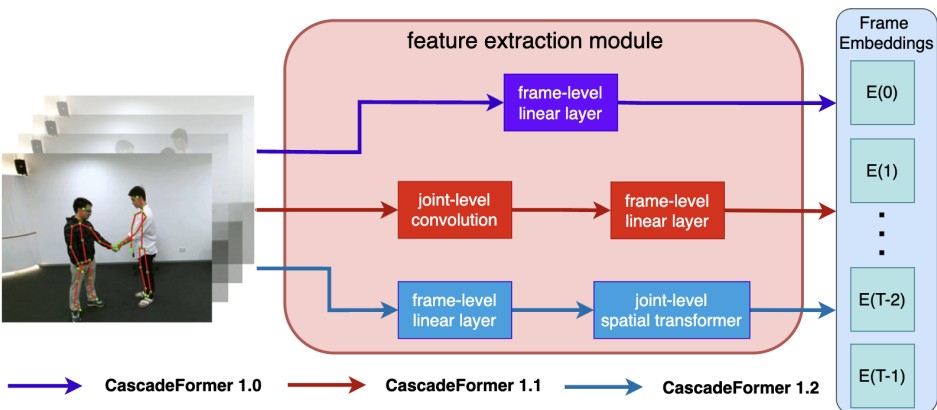

Figure 3: **Feature extraction module in CascadeFormer.** All variants convert input skeleton sequences into frame-level embeddings for downstream temporal modeling. CascadeFormer 1.0 (purple) applies a simple frame-level linear projection. CascadeFormer 1.1 (red) enhances this by first applying a joint-level 1D convolution to capture spatial locality before linear projection. CascadeFormer 1.2 (blue) constructs joint-level embeddings via a linear layer, refines them using a joint-level spatial transformer, and aggregates the outputs into frame-level embeddings.

As illustrated in Figure 3, input skeleton sequences are first passed through a feature extraction module to obtain preliminary frame-level embeddings E, which are then processed by the transformer backbone (T1). We design a family of three CascadeFormer variants, each differing in how frame embeddings are extracted:

**CascadeFormer 1.0** This baseline variant employs a simple linear projection to generate frame embeddings directly from the input skeleton. Formally, given the input skeleton tensor $X \in \mathbb{R}^{B \times C \times T \times J}$, where B is the batch size, C the coordinate dimension, T the number of frames, and J the number of joints, the linear projection outputs: $E \in \mathbb{R}^{B \times T \times \text{embed\_dim}}$.

**CascadeFormer 1.1** To better capture spatial locality among joints, this variant prepends a lightweight convolutional module prior to linear projection. The input skeletons $X \in \mathbb{R}^{B \times C \times T \times J}$ are reshaped in batch and temporal dimensions into $X \in \mathbb{R}^{(B \cdot T) \times C \times J}$, and then passed through a 1D convolutional layer: $X_{\text{convoluted}} \in \mathbb{R}^{(B \cdot T) \times C \times J}$. The resulting activations are passed into the same linear projection as in CascadeFormer 1.0 to yield frame-level embeddings: $E \in \mathbb{R}^{B \times T \times \text{embed\_dim}}$.

**CascadeFormer 1.2** In this variant, we first construct embeddings for each individual joint using a linear layer, producing: $E_{\text{joint}} \in \mathbb{R}^{(B \cdot T) \times J \times (\text{embed\_dim}/J)}$ A single layer spatial transformer (denoted as ST) is then applied to the joint embeddings. $E_{\text{joint\_ST}} \in \mathbb{R}^{(B \cdot T) \times J \times (\text{embed\_dim}/J)}$. Finally, we aggregate joint-level output into full frame-level embeddings $E \in \mathbb{R}^{B \times T \times \text{embed\_dim}}$. Note that for CascadeFormer 1.2, the total embedding dimension embed\_dim must be divisible by the number of joints J to allow for an even allocation across the joints. All three variants differ only in their feature extraction mechanisms; the remainder of the architecture, including masked pre-training (Figure 1) and cascading finetuning (Figure 2)—remains shared across all models.

## 5 EXPERIMENT SETUP

In this section, we introduce the datasets used for evaluation, followed by dataset-specific preprocessing steps and training configurations.

## 5.1 DATASETS

We conduct a comprehensive evaluation of CascadeFormer on three widely-used benchmark datasets for human action recognition. **Penn Action** Zhang et al. (2013) dataset contains 2,326 video clips that span 15 human action classes. Each frame is annotated with 13 human joints in a 2D space. The dataset provides a standard 50/50 train/test split, ensuring mutually exclusive samples in both sets. **N-UCLA** wang et al. (2014) comprises 1,494 video clips categorized into 12 action classes. Skeletons are annotated with 20 joints in a 3D space. The dataset is collected from three camera viewpoints: two are used for training, and the third is reserved for evaluation, following the standard cross-view evaluation protocol. **NTU-RGB+D 60** Shahroudy et al. (2016), a large-scale dataset includes 56,880 video samples covering 60 action classes, with each skeleton containing 25 joints in 3D space. NTU-RGB+D 60 offers two evaluation protocols: cross-subject (where subjects are split between training and testing) and cross-view (where specific camera angles are held out for testing). We adopted both protocols in our experiments.

## 5.2 DATA PREPROCESSING

Due to differences in scale, format, and split protocols across datasets, we apply dataset-specific preprocessing strategies. For **Penn Action**, we take advantage of the visibility flags provided with the annotations and remove occluded skeletons. Given the relatively small size of the dataset, we pad each video sequence within a batch to match the longest sequence in that batch. For **N-UCLA**, following the data pre-processing strategy in SkateFormer Do & Kim (2024), we virtually repeat the dataset multiple times to increase the diversity of the training. Each skeleton is normalized by centering it at the hip joint and applying scale normalization. Data augmentation techniques such as random rotation, scaling, and joint/axis dropping are applied. To avoid computational overhead from batch-level padding, we randomly sampled a fixed length of 64 frames per sequence. For **NTU-RGB+D 60**, we adopt the preprocessing pipeline from CTR-GCN Chen et al. (2021), which includes random rotation augmentation. To handle variable-length sequences, we uniformly sample 64 frames around the temporal center of each clip.

## 5.3 TRAINING SETUP

All models are implemented in PyTorch Paszke et al. (2019) and trained on a single NVIDIA GeForce RTX 3090 GPU. We adopt dataset-specific hyperparameters to ensure optimal performance across different datasets. For **Penn Action and N-UCLA**, we use AdamW Loshchilov & Hutter (2019) as the optimizer. For masked pre-training, the learning rate is fixed at $1.0 \times 10^{-4}$. During the cascading fine-tuning stage, we reduce the learning rate to $1.0 \times 10^{-5}$ and apply cosine annealing scheduling Loshchilov & Hutter (2017). For **NTU-RGB+D 60**, masked pretraining is performed with a constant learning rate of $1.0 \times 10^{-4}$ and cosine annealing. During cascading fine-tuning, we switch to SGD Ruder (2017) as the optimizer. For cross-subject evaluation, we use a base learning rate of $3.0 \times 10^{-5}$, while for cross-view evaluation, we set it to $1.0 \times 10^{-4}$. The number of epochs for both pre-training and fine-tuning varies across the three CascadeFormer variants.

# 6 RESULTS

## 6.1 DATASET-WISE EVALUATION

We present the performance of the CascadeFormer variants in three benchmark datasets in Table 1. In general, each variant demonstrates competitive performance, with certain variants better suited to specific datasets. In the Penn Action dataset, **CascadeFormer 1.0** achieves the highest accuracy at 94.66%, while both **1.1** and **1.2** closely follow at 94.10%. On the **N-UCLA** dataset, **Cascade-Former 1.1** leads with an accuracy of 91.16%, while variants **1.2** and **1.0** reach 90.73% and 89.66%, respectively. For the large-scale NTU RGB+D 60 dataset , which includes two evaluation protocols - cross-subject (CS) and cross-view (CV)—**CascadeFormer 1.0** again outperforms the others with **81.01%** (CS) and **88.17%** (CV). Variant 1.2 closely follows, while 1.1 lags slightly behind on both splits. These results suggest that, perhaps surprisingly, the simplest feature extraction design (Cas-cadeFormer 1.0 with a linear frame encoder) can be as effective, if not more, than more complex

designs involving convolutions or spatial transformers. All model checkpoints used in Table 1 are publicly released via HuggingFace for further analysis and reproducibility.

| Model Variant | Penn Action | N-UCLA | NTU60/CS | NTU60/CV |
|---|---|---|---|---|
| *accuracy* | | | | |
| CascadeFormer 1.0 | **94.66%** | 89.66% | **81.01%** | **88.17%** |
| CascadeFormer 1.1 | 94.10% | **91.16%** | 79.62% | 86.86% |
| CascadeFormer 1.2 | 94.10% | 90.73% | 80.48% | 87.24% |
| *precision* | | | | |
| CascadeFormer 1.0 | **94.36%** | 90.72% | **80.86%** | **88.34%** |
| CascadeFormer 1.1 | 93.63% | **91.63%** | 79.53% | 87.01% |
| CascadeFormer 1.2 | 93.94% | 91.57% | 80.33% | 87.46% |
| *F1 score* | | | | |
| CascadeFormer 1.0 | **94.18%** | 89.59% | **80.82%** | **88.18%** |
| CascadeFormer 1.1 | 93.66% | **91.12%** | 79.40% | 86.84% |
| CascadeFormer 1.2 | 93.88% | 90.75% | 80.28% | 87.26% |

Table 1: **Overall accuracy, precision, and F1 score evaluation results of CascadeFormer variants on three datasets.** All checkpoints are open-sourced for reproducibility.

## 6.2 MULTI-PERSON ACTION ANALYSIS

Unlike Penn Action and N-UCLA, which feature only single-person actions, NTU RGB+D 60 includes 11 classes involving interactions between two individuals (e.g., hugging, handshaking, pushing). Following the recommendation of the NTU authors Shahroudy et al. (2016), we retain the person with the greatest variance in motion for all actions. As shown in Table 2, **CascadeFormer 1.0** achieves 81.81% accuracy in two-person actions (CS split), slightly higher than the 80.82% obtained in single-person actions. This suggests strong generalization across subjects for interaction-based actions, despite a smaller number of classes. However, on the CV split, the model performs better on single-person actions (88.92%) than on two-person actions (84.86%), indicating that generalizing interactions across varying camera viewpoints remains more challenging than across different individuals. These findings reveal that cross-view generalization of social interactions may benefit from additional structural modeling of inter-person dynamics, which could be explored in future work.

| Action Type | NTU60/CS accuracy | NTU60/CV accuracy |
|---|---|---|
| Single-person | 80.82% | **88.92%** |
| Two-persons | **81.81%** | 84.86% |
| Overall | 81.01% | 88.17% |

Table 2: **Performance of CascadeFormer 1.0 on multi-person actions in NTU RGB+D 60.** Accuracy is broken down into single-person actions and two-person interactions under both cross-subject and cross-view splits.

## 7 PERFORMANCE COMPARISON

In this section, we compare the performance of the three CascadeFormer variants across different datasets with other representative models in skeleton-based action recognition. Table 3 compares the model accuracy in the Penn Action dataset Zhang et al. (2013). AOG Nie et al. (2015), a hierarchical graph model, reaches 85.5% accuracy. HDM-BG Zhao et al. (2019), a probabilistic model within a Bayesian framework, achieves 93.4%. All three CascadeFormer variants exceed 94% accuracy, with **CascadeFormer 1.0** achieving the best performance at **94.66%**. These results demonstrate the effectiveness of CascadeFormer on compact 2D skeleton datasets like Penn Action. Table 4 shows the results on the N-UCLA dataset wang et al. (2014). ESV Liu et al. (2017), a CNN-based visualization method, obtains 86.09%, while Ensemble TS-LSTM Lee et al. (2017), an RNN-based model, reaches 89.22%. CascadeFormer 1.0 already exceeds this with 89.66%, and 1.1 and 1.2

| Model | Penn Action accuracy |
|---|---|
| AOG | 85.5% |
| HDM-BG | 93.4% |
| CascadeFormer 1.0 | **94.66%** |
| CascadeFormer 1.1 | 94.10% |
| CascadeFormer 1.2 | 94.10% |

Table 3: **Comparison on Penn Action Zhang et al. (2013).** AOG Nie et al. (2015) is a hierarchical graph-based model. HDM-BG Zhao et al. (2019) is a probabilistic Bayesian approach. Cascade-Former variants achieve top accuracy, with 1.0 reaching the highest.

| Model | N-UCLA acc |
|---|---|
| ESV | 86.09% |
| Ensemble TS-LSTM | 89.22% |
| CascadeFormer 1.0 | 89.66% |
| CascadeFormer 1.1 | **91.16%** |
| CascadeFormer 1.2 | 90.73% |

Table 4: **Comparison on N-UCLA wang et al. (2014).** ESV Liu et al. (2017) is a CNN-based visualization method. TS-LSTM Lee et al. (2017) is a recurrent model ensemble. CascadeFormer 1.1 achieves the highest accuracy.

further improve to **91.16%** and 90.73%, respectively. These results confirm that CascadeFormer performs robustly on small-scale 3D skeleton datasets as well.

| Model | N-UCLA accuracy | Model | NTU60/CS accuracy |
|---|---|---|---|
| ESV | 86.09% | ST-LSTM | 69.2% |
| Ensemble TS-LSTM | 89.22% | ST-GCN | **81.5%** |
| CascadeFormer 1.0 | 89.66% | CascadeFormer 1.0 | 81.01% |
| CascadeFormer 1.1 | **91.16%** | CascadeFormer 1.1 | 79.62% |
| CascadeFormer 1.2 | 90.73% | CascadeFormer 1.2 | 80.48% |

Table 5: **Comparison on NTU RGB+D 60 (both Cross-Subject and Cross-View).** ST-LSTMLiu et al. (2016) uses recurrent modeling. ST-GCN Yan et al. (2018) adopts graph convolutions. Cas-cadeFormer 1.0 achieves performance comparable to ST-GCN without graph structures.

On the NTU RGB+D 60 dataset, we evaluate both cross-subject (CS) and cross-view (CV) splits. As shown in Table 5, ST-LSTM Liu et al. (2016) achieves 69.2% (CS) and 77.7% (CV), while ST-GCN Yan et al. (2018), a graph-based method, improves performance to 81.5% and 88.3%, respectively. CascadeFormer 1.0 delivers comparable performance—**81.01%** (CS) and **88.17%** (CV)—despite not using any explicit graph structure. CascadeFormer 1.2 follows closely behind, achieving 80.48% and 87.24%. These results validate CascadeFormer scalability and robustness in complex large-scale 3D datasets. Even without spatial graphs, our model competes with state-of-the-art graph convolutional approaches.

## 8 ABLATION HIGHLIGHTS

This section presents two key ablation studies that highlight the effectiveness of our proposed design choices. Additional ablation results and comprehensive analyses can be found in the appendix.

### 8.1 NECESSITY OF STRONG PRETRAINING

In this ablation study, we examine the necessity of employing a strong transformer backbone (T1) through masked pre-training. Its advantages of lie in the ability to decouple generic representation learning from task-specific adaptation. Specifically, we conducted a case study on **CascadeFormer**

**1.0** using the cross-subject split of the NTU RGB+D 60 dataset, a large-scale and complex benchmark that is well suited to reveal performance differences under varying pre-training strengths. We pre-train the T1 backbone for 1, 50, 100, and 200 epochs, followed by 100 epochs of fine-tuning in all settings to ensure fair comparison. A weak backbone pretrained for only 1 epoch yields an accuracy of 76.38%; A medium backbone pretrained for 50 epochs leads to 80.06% accuracy. Extending pretraining to 100 epochs results in a further improvement to 81.01%. Although training for 200 epochs yields a marginal increase to 81.09%, the additional computational cost outweighs the negligible performance gain. Therefore, we adopted 100 epochs of masked pretraining for all subsequent experiments. This study confirms that strong backbone, achieved by sufficient masked pre-training, is necessary and beneficial for robust performance in challenging action recognition tasks.

| # Epochs of Pretraining | # Epochs of Pretraining | NTU60/CS accuracy |
|---|---|---|
| 1 epoch | 100 epochs | 76.38% |
| 50 epoch | 100 epochs | 80.06% |
| 100 epochs | 100 epochs | **81.01%** |
| 200 epochs | 100 epochs | 81.09% |

Table 6: **Effect of pretraining duration on NTU RGB+D 60 (cross-subject) performance.** We report accuracy on the cross-subject split after pretraining **CascadeFormer 1.0** for 1, 50, 100, and 200 epochs. Performance improves significantly with longer pretraining, but plateaus after 100 epochs, indicating diminishing returns beyond this point.

## 8.2 PRETRAINING STRATEGY

In this ablation study, we further explore two alternative pretraining strategies using **CascadeFormer 1.0** on the Penn Action dataset Zhang et al. (2013), with results summarized in Table 7. Our proposed strategy applies random masking to 30% of joints across the entire video sequence, and optimizes a reconstruction loss in order to learn the most robust skeleton representations. This method achieves a high accuracy of 94.66% on Penn Action. As a comparison, we evaluated a frame-level masking strategy, where 30% of entire frames are randomly masked. This results in a lower accuracy of 89.98%, indicating that reconstructing full frames is significantly more challenging, less stable, and therefore less reliable for the transformer backbone than reconstructing scattered joint positions. We also investigate a **non-masked** variant in which the full skeleton sequence is reconstructed without any masking. This strategy achieves 91.10% accuracy, suggesting that while full reconstruction may be effective, it can lead to overfitting on smaller datasets, such as Penn Action, due to hard skeleton memorization without any regularization. Based on these observations, we adopt the random joint masking strategy as our default pretraining approach for all subsequent experiments.

| Pretraining Strategy | Penn Action accuracy |
|---|---|
| random joint masking | **94.66%** |
| random frame masking | 89.98% |
| regular reconstruction | 91.10% |

Table 7: **Effect of pretraining strategy on Penn Action performance.** We compare three pretraining objectives for **CascadeFormer 1.0**: random joint masking, random frame masking, and regular (unmasked) reconstruction. Random joint masking yields the best downstream performance, highlighting the benefit of fine-grained spatial masking.

## 9 CONCLUSION

In this paper, we introduced CascadeFormer, a transformer-based model for skeleton-based action recognition with masked pretraining and cascading finetuning. Evaluated on Penn Action, N-UCLA, and NTU RGB+D 60, CascadeFormer consistently achieved strong performance on spatial modalities and dataset scales. Our results demonstrate the effectiveness of unified transformer pipelines for generalizable action recognition. Future directions include scaling to longer sequences, incorporating graph-aware modules, and extending to multimodal settings with RGB or depth inputs.

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

# A APPENDIX

## A.1 BROADER IMPACTS

The empirical results across all three datasets provide a couple of insights into the design and performance of CascadeFormer: First, the effectiveness of the cascading fine-tuning strategy is consistent across model variants and datasets. CascadeFormer variants demonstrate clear improvements over strong baselines in both 2D and 3D skeleton-based action recognition tasks. Second, CascadeFormer demonstrates competitive scalability to large-scale and complex datasets. On NTU RGB+D 60, despite the absence of an explicit graph structure, CascadeFormer achieves competitive accuracy performance. This suggests that transformer-based architectures, when paired with effective pretraining strategies, can be comparable to graph-based methods in modeling human motion dynamics.

## A.2 MORE ABLATION STUDIES

We conduct further ablation studies to examine the effects of input representations, decoder architectures, backbone freezing strategies, and masking ratios.

### A.2.1 INPUT DATA REPRESENTATION

We investigate alternative input data representations beyond the original joint coordinates using **CascadeFormer 1.0**, as shown in Table 8. Surprisingly, the use of raw joint coordinates yields the highest accuracy of **94.66%** on the Penn Action dataset. To explore the effectiveness of bone-based representations, we develop three variants. The first approach constructs each bone by subtracting the coordinates of one joint from its adjacent joint, resulting in an accuracy of 92.32%. The second method concatenates the coordinates of two consecutive joints, achieving 93.16% accuracy. The third approach linearly parameterizes each bone segment using its slope and intercept, which attains 93.91% accuracy. Overall, although all bone-based variants perform competitively, the original joint representation proves to be the most effective for our model on this task.

| Data Representation | Accuracy |
|---|---|
| Joints | **94.66%** |
| Bones (Subtraction) | 92.32% |
| Bones (Concatenation) | 93.16% |
| Bones (Parameterization) | 93.91% |

Table 8: **Comparison of different input data representations on Penn Action**. Using the original joint coordinates achieves the highest accuracy (94.66%), outperforming all three alternative bone-based variants.

### A.2.2 DECODER ARCHITECTURE

We conduct an ablation study to compare alternative decoder architectures for masked pretraining on the *CascadeFormer 1.0* using the Penn Action dataset. As shown in Table 9, our default choice—a simple linear layer to reconstruct masked joints—achieves the highest accuracy of **94.66%**. We then evaluate an MLP decoder composed of two linear layers with a ReLU activation in between, which yields a reduced accuracy of 92.51%. Finally, we test an MLP decoder with a residual connection to facilitate gradient flow, resulting in an even lower accuracy of 91.20%. These results suggest that the linear decoder not only provides the most effective reconstruction but also generalizes better, likely due to its lower risk of overfitting on small-scale datasets. Based on this finding, we adopt the linear decoder for all experiments throughout this work.

| Decoder Architecture | Accuracy |
|---|---|
| linear | **94.66%** |
| MLP | 92.51% |
| MLP + residual | 91.20% |

Table 9: **Comparison of decoder architectures during masked pretraining on Penn Action**. A simple linear decoder outperforms both MLP and MLP with residual connection, indicating that increased decoder complexity may lead to overfitting on smaller datasets.

### A.2.3   BACKBONE FREEZING DECISION

In this ablation study, we examine the impact of parameter-freezing strategies for the transformer backbone during the cascading finetuning stage. Using *CascadeFormer 1.0* on the Penn Action dataset, we present our findings in Table 10. Our primary approach involves fully finetuning the entire backbone, allowing all transformer parameters to be updated during training. This strategy yields the best performance with an accuracy of **94.66%**. In contrast, freezing the entire backbone results in a significant drop in accuracy to 85.11%. We also explore partial finetuning by training only the final transformer layer, which achieves 88.39% accuracy. These results suggest that full backbone finetuning is crucial for effective downstream adaptation in action recognition. Consequently, we adopt full backbone finetuning for all experiments.

| Backbone Freezing Decision | Accuracy |
|---|---|
| fully finetune | **94.66%** |
| fully freeze | 85.11% |
| finetune the last layer | 88.39% |

Table 10: **Effect of different backbone freezing strategies during cascading finetuning on Penn Action.** Fully finetuning the transformer backbone yields the highest accuracy, while freezing all layers significantly degrades performance.

### A.2.4   MASKING RATIOS

The ablation study on masking ratios highlights the importance of selecting an appropriate level of input corruption during masked pretraining. As shown in Table 11, a 30% masking ratio achieves the highest accuracy of 94.66%, substantially outperforming both lower (20%) and higher (40%) ratios. This indicates that too little masking may not provide sufficient pretraining signal to encourage robust feature learning, while excessive masking reduces the availability of contextual information and hinders accurate reconstruction. The results suggest that an intermediate masking ratio strikes the optimal balance between learning meaningful representations and maintaining adequate context for recovery, consistent with findings in related masked modeling literature.

| Masking Ratio | Accuracy |
|---|---|
| 30% | **94.66%** |
| 20% | 89.04% |
| 40% | 88.20% |

Table 11: **Effect of masking ratios during pretraining.** An intermediate ratio of 30% yields the best downstream accuracy, while both lower (20%) and higher (40%) ratios degrade performance.

