# OpenReview forum: "CascadeFormer: A Family of Two-stage Cascading Transformers for Skeleton-based Human Action Recognition"
_ICLR.cc/2026/Conference — ICLR 2026 Conference Withdrawn Submission_

### Official Review · Reviewer_sVLn · 2025-10-29

**Soundness:** 1
**Presentation:** 2
**Contribution:** 1
**Rating:** 2
**Confidence:** 4

**Summary:**

The authors propose the CascadeFormer, a family of two-stage cascading transformers for skeleton-based human action recognition. The framework consists of a masked pretraining stage to learn generalizable skeleton representations, followed by a cascading fine-tuning stage tailored for discriminative action classification. Experiments are conducted on the Penn Action, N-UCLA, and NTU RGB+D 60 datasets to validate the effectiveness of the proposed models.

**Strengths:**

Strengths:
(1) A large-scale pre-trained transformer model for skeleton representations is desirable for the community of human action recognition.

(2) It is interesting to explore masked pre-training method for skeleton representation learning.

**Weaknesses:**

(1) The authors lack a thorough understanding of the existing literature. Several pre-training methods for skeleton-based action recognition have been proposed, such as MotionBert [*], SkeletonMAE [**], and MAMP [***], etc. These methods also utilize Masked Sequence Modeling Tasks for pre-training to seek a unified representation of human skeletal actions. It is recommended that the authors review the latest related work [****] for a comprehensive understanding on recent advanced methods.

- [*] Zhu, X. Ma, Z. Liu, L. Liu, W. Wu, and Y. Wang, “Motionbert: Aunified perspective on learning human motion representations,” In CVPR, 2023.

- [**] H. Yan, Y. Liu, Y. Wei, Z. Li, G. Li, and L. Lin, “SkeletonMAE: graph based masked autoencoder for skeleton sequence pre-training,” In CVPR, 2023.

- [***] Y. Mao, J. Deng, W. Zhou, Y. Fang, W. Ouyang, and H. Li, “Masked motion predictors are strong 3d action representation learners,” In CVPR, 2023

- [****]H. Wang, W. Weng, J. Wang, F. Zhao, G. Xie, X. Geng, and L. Wang, “Foundation Model for Skeleton-Based Human Action Understanding,” In TPAMI, 2025.

(2) The experimental results also do not verify the effectiveness of the proposed method CascadeFormer, especially lacking a performance comparison with the state-of-the-art (SOTA) methods on public benchmarks. For instance, the SkateFormer [Do & Kim (2024)]  model, which is also based on the Transformer architecture, has achieved over 97% performance in the Cross-View task on the NTU RGB+D 60 dataset, whereas CascadeFormer only reaches around 88%.

From both perspectives mentioned above, this submission neither presents sufficiently innovative pre-training ideas or methods nor brings improvements in model performance. It is recommended for rejection.

**Questions:**

In the section of TRAINING SETUP,  the masked pretraining is performed within a dataset-specific strategy.  Athough the skeleton formats are different across datasets, it is more desirable for learning an unified representation across different datasets. So why not perform the maksed pre-training over the training sets of all the three datasets, or using a more large scale skeleton dataset, such as Kinect 700 ？ Some preprocessing can be performed in a dataset specific strategy for transform them into a unified skeleton format.

---

### Official Review · Reviewer_mRBP · 2025-10-30

**Soundness:** 2
**Presentation:** 2
**Contribution:** 1
**Rating:** 2
**Confidence:** 4

**Summary:**

This paper proposes CascadeFormer, a two-stage cascading transformer framework for skeleton-based human action recognition. The approach consists of a masked pretraining stage to learn generalizable skeleton representations, followed by a cascading fine-tuning stage with an additional task-specific transformer (T2) for action classification. Three model variants are proposed (CascadeFormer 1.0, 1.1, 1.2) that differ in their feature extraction modules. The method is evaluated on three benchmark datasets: Penn Action, N-UCLA, and NTU RGB+D 60, achieving competitive results.

**Strengths:**

1) The integration of masked pretraining with vanilla transformers adapts successful strategies from language and vision domains to skeleton data, potentially improving generalization over traditional end-to-end training approaches.
2) The framework is modular with three variants exploring different feature extraction approaches.

**Weaknesses:**

1) While the approach is solid, the novelty is incremental—transformers and masked pretraining are well-established in vision/language, and skeleton-specific adaptations feel straightforward without deep innovation in attention mechanisms or handling multi-person dynamics.
2) On NTU RGB+D 60, CascadeFormer achieves only 81.01% (CS) and 88.17% (CV), significantly below current SOTA models, indicating limited competitiveness on large-scale datasets due to insufficient capture of complex action dynamics.
Additionally, the paper compares only with early models (e.g., AOG, HDM-BG) while ignoring recent advanced transformer-based methods (e.g., SkateFormer, CTR-GCN), making the claimed advantages less convincing.
3) No Efficiency Discussion: Cascading two transformers (T1 and T2) increases parameters and computation, yet the paper provides no model size, inference time, or comparison with lightweight GCNs, ignoring real-world deployment concerns.

**Questions:**

Please refer to the weakness.

---

### Official Review · Reviewer_Numk · 2025-10-30

**Soundness:** 2
**Presentation:** 2
**Contribution:** 2
**Rating:** 4
**Confidence:** 4

**Summary:**

This paper introduces the cascadeformer framework which is a two-stage cascading transformer, with masked pretraining strategy for skeleton-based action recognition.

They introduce some variants of the model, and the evaluations are conducted on 3 benchmark datasets.

The evaluations, analyses and comparisons show improved performance.

**Strengths:**

+ This paper is written in a quite clean way.

+ The proposed cascadeformer model shows improvements compared to some existing methods.

+ The evaluations on their variants are clear, and show the benefits of different versions.

**Weaknesses:**

- It is unclear what motivates this work, why motivate the use of masked pretraining and on what dataset for such pretraining concept. Regarding the use of transformer, why and how, what motivate the use of two-stage cascadeformer? In intro, the last two listed contributions do not sound like contributions.

- Related work section, the authors do not outline how this work differs from existing works. It would be better to also review some works that use eg hypergraph-based models rather than GCNs etc for skeletons, such as [A]. How this work differs from existing works that also use transformers?

[A] L Wang and P Koniusz. 3Mformer: Multi-order Multi-mode Transformer for Skeletal Action Recognition. Proceedings of the IEEE/CVF Conference on Computer Vision and Pattern Recognition (CVPR). 2023.

- The method section appears to be unclear. (i) notations regarding maths symbols and operations could be standardised eg what are scalars, vectors and matrices etc. (ii) method section need to be revised to be clearly and logical flow. Model architectures are unclear and vague, the authors do not provide any descriptions on these such as modules, blocks, how many layers, how to stack those properly etc. (iii) some maths symbols and operations are not being clearly explained before using them, also equations are not being properly numbered.

- The datasets used in this paper appear to be a bit small scale and old-fashioned. On the other hand, the performance reported in the literature for Penn Action and N-UCLA tend to be saturated. The authors should explore the use of new large-scale / challenging datasets such as [B] in evaluation.

[B] Y Liu, J Yang, M Perera, P Ji, D Kim, M Xu, T Wang, S Anwar, T Gedeon, L Wang, Z Qin. Representation-Centric Survey of Skeletal Action Recognition and the ANUBIS Benchmark. arXiv preprint arXiv:2205.02071, 2025.

- The evaluations and comparisons tend to be limited. The authors should compare with newest/recent state-of-the-art methods to show the improvements of the proposed method. The current methods being used in comparison such as in Table 3-5 tend to be quite old (eg, from more than 10 years ago), also the selected methods lack of justification, and it would be better to compare with recent works. Another concern is that why different methods are being used in comparison across different datasets, it would be better to choose the same set of existing methods in comparison to show the effectiveness and robustness of the proposed model.

- Ablation studies tend to be limited. Currently most results are presented in the form of tables, it is suggested to use a diverse set of representations to show the evaluations eg plots or figures etc.

- The organisation of this paper needs significant improvements.

- Most references are cited in the form of arXiv, it is suggested to update to be their published versions.

- Lack of declaration on the use of LLM. It would be better to have a section discussing the limitations of this work and outlining some future research directions.

**Questions:**

Refer to weaknesses and also some extra questions below.

- Some of the paragraphs read not very smooth, such as Line 44-48, it seems not very well connected. Also in Line 44, why trained in an “end-to-end” manner is not good (the authors use “however” here)?

- What is two-stage, and how the “cascading transformer” is applied?

- It is unclear how “remove occluded skeletons” is being performed (Line 288-289).

- why table 6 has two columns of “#epochs of pretraining”?

---

### Official Review · Reviewer_VQwc · 2025-10-30

**Soundness:** 2
**Presentation:** 2
**Contribution:** 2
**Rating:** 2
**Confidence:** 5

**Summary:**

This paper introduces CascadeFormer for skeleton-based action recognition. Unlike traditional graph convolution networks or end-to-end frameworks, CascadeFormer decouples representation learning and task-specific adaptation. Firstly, pre-training is performed by masking some joints and reconstructing them. Secondly, the model is trained for action recognition using a transformer-based model. The proposed method is evaluated on Penn Action, N-UCLA, and NTU RGB+D 60 datasets.

**Strengths:**

1. The two stage training is interesting in skeleton-based action recognition.
2. The paper is easy to follow.
3. Code is provided for reproducibility

**Weaknesses:**

1. The motivation of using masked pretraining is questionable. I understand it helps in language modeling and image representation learning. It is unclear why reconstructing joint positions can help the generalization of action recognition. And there are no experiments to demonstrate this.
2. The feature extraction modules seems some layer tuning. The method part seems like an ablation technical report
3. No comparison with state-of-the-art methods.
4. Missing results on larger datasets such as NTU-RG+D 120.

**Questions:**

When doing the finetuning, are all joints used? If so, for such datasets, is the pretraining effective?

---

### Note · Authors · 2025-11-18

I have read and agree with the venue's withdrawal policy on behalf of myself and my co-authors.